# Assembly and Comparison of *Ca.* Neoehrlichia mikurensis Genomes

**DOI:** 10.3390/microorganisms10061134

**Published:** 2022-05-31

**Authors:** Tal Azagi, Ron P. Dirks, Elena S. Yebra-Pimentel, Peter J. Schaap, Jasper J. Koehorst, Helen J. Esser, Hein Sprong

**Affiliations:** 1Centre for Infectious Diseases Research, National Institute for Public Health and the Environment, 3720 BA Bilthoven, The Netherlands; hein.sprong@rivm.nl; 2Future Genomics Technologies BV, 2333 BE Leiden, The Netherlands; dirks@futuregenomics.tech (R.P.D.); yebra-pimentel@futuregenomics.tech (E.S.Y.-P.); 3Laboratory of Systems and Synthetic Biology, Wageningen University & Research, 6708 PB Wageningen, The Netherlands; peter.schaap@wur.nl (P.J.S.); jasper.koehorst@wur.nl (J.J.K.); 4UNLOCK, Wageningen University, 6708 PB Wageningen, The Netherlands; 5Wildlife Ecology & Conservation Group, Wageningen University, 6708 PB Wageningen, The Netherlands; helen.esser@wur.nl

**Keywords:** tick-borne pathogens, *Ixodes ricinus*, Nanopore sequencing, rickettsiales, Anaplasmataceae

## Abstract

*Ca.* Neoehrlichia mikurensis is widely prevalent in *I. ricinus* across Europe and has been associated with human disease. However, diagnostic modalities are limited, and much is still unknown about its biology. Here, we present the first complete *Ca.* Neoehrlichia mikurensis genomes directly derived from wildlife reservoir host tissues, using both long- and short-read sequencing technologies. This pragmatic approach provides an alternative to obtaining sufficient material from clinical cases, a difficult task for emerging infectious diseases, and to expensive and challenging bacterial isolation and culture methods. Both genomes exhibit a larger chromosome than the currently available *Ca.* Neoehrlichia mikurensis genomes and expand the ability to find new targets for the development of supportive laboratory diagnostics in the future. Moreover, this method could be utilized for other tick-borne pathogens that are difficult to culture.

## 1. Introduction

*Ixodes ricinus* is the most abundant and widespread tick species in Europe [1,2] and transmits multiple pathogens of medical and veterinary concern [3]. Two well-established tick-borne diseases, Lyme borreliosis and tick-borne encephalitis, are frequently reported in Europe, and several studies have indicated a rise in their incidences and spread over the last decades [4,5]. *Ixodes ricinus* also transmits *Ca.* Neoehrlichia mikurensis, which has been found all across Europe except for the United Kingdom [6,7,8]. The number of studies describing human infections involving *Ca.* N. mikurensis is accumulating, but whether these infections result in human disease has not been fully demonstrated [9,10].

One of the major obstacles to investigating how often and under what conditions *Ca.* N. mikurensis causes an infectious disease in humans, is its unequivocal detection in larger cohorts of patients with noncharacteristic disease symptoms [11,12] or in persons with a (recent) tick bite [13,14,15]. In other words, supportive laboratory diagnostics to detect and identify *Ca.* N. mikurensis infections are currently limited or reserved to research laboratories. Most importantly, although cultivation of *Ca.* N. mikurensis has been described in the literature recently [16], it turns out to be quite difficult, even in dedicated laboratories [17,18]. Once one or more *Ca.* N. mikurensis cultures are generally available, it will become possible to develop more specific and sensitive diagnostic modalities, for example serological tests, which will undoubtedly improve the abilities to detect (endured) infection with *Ca.* N. mikurensis in clinical practices as well as in epidemiological studies. Recently, genomic information of three *Ca.* N. mikurensis isolates from Swedish patients became available, giving new insights into its genetic make-up [18].

Since genetic material of emerging tick-borne pathogens from confirmed clinical cases is hard to acquire, our aim was to test whether natural reservoir hosts, a more available source of tissues with high bacteremia, may be utilized as an alternative source for whole genome sequencing. Whole genome sequencing of bacterial and viral pathogens is increasingly used for serotyping and outbreak management [19,20,21]. Previous studies have shown the importance of high-quality DNA as a starting point [22]. Moreover, the use of hybrid assembly strategies, combining long- and short-read sequencing, is recommended for more complete and accurate reference genome assemblies [19,22,23,24,25], especially when repetitive elements are expected. This approach has already been used for other members of the Anaplasmataceae in order to assemble high-resolution whole genomes [26,27]. The long reads provide reliable contigs allowing for repeat regions and complex sequences to be structurally correct, and polishing with accurate short reads corrects errors in the assembly [22,23,28]. Once standardized, an approach in which DNA is extracted following bacterial enrichment, and then sequenced with a hybrid approach, could be used to obtain fully resolved genomes from both host and patient samples.

In this study, we show that two complete and circular *Ca.* N. mikurensis genomes could be obtained without the need for time- and resource-consuming isolation and culture. These full genomes were directly derived from spleen samples of bank voles (*Myodes glareolus*) from the Netherlands, using a hybrid assembly approach combining PromethION and Illumina NovaSeq 6000 sequencing. Both genomes exhibit a larger chromosome than the currently available genomes and expand the ability to find new targets for the development of diagnostics in future studies.

## 2. Materials and Methods

### 2.1. Sample Collection and Storage

Rodents were collected in various locations in the Netherlands between August and October 2018 (Appendix A). Rodents were trapped using Heslinga live traps that were filled with hay and baited with a mixture of grains, carrots, and mealworms. Captured rodents were transported to the laboratory facility where they were anesthetized using isoflurane, after which the animals were euthanized by cervical dislocation. Species identification was performed both morphologically and molecularly [29]. For this study, *Myodes glareolus* were dissected, and spleen samples were taken and stored at −80 °C. All handling procedures were approved by the Animal Experiments Committee of Wageningen University (2017.W-0049.003 and 2017.W-0049.005) and by the Netherlands Ministry of Economic Affairs (FF/75A/2015/014).

### 2.2. DNA Extraction, Pathogen Detection, and Enrichment

DNA from spleen samples was extracted using the Qiagen DNeasy Blood & Tissue Kit according to the manufacturer’s manual (Qiagen, 2006, Hilden, Germany), and screened for the presence of *Ca.* N. mikurensis DNA using a qPCR targeting a fragment of the *groEL* gene (Appendix A). DNA from two qPCR-positive spleen samples from a male bank vole (samples 18-2804 and 18-2837, Appendix A) was extracted, this time using the Invitrogen genomic DNA mini kit (Qiagen, Germany) in order to ensure higher genomic DNA yield for NGS. Microbial DNA enrichment was achieved by selective binding and removal of the CpG-methylated host DNA using the NEBNext^®^ Microbiome DNA Enrichment Kit (NEB, Frankfurt am Main, Germany). DNA quality was measured via electrophoresis in Genomic DNA ScreenTape on an Agilent 4200 TapeStation System (Agilent Technologies Netherlands BV, Amstelveen, The Netherlands), and DNA quantity was measured using Qubit dsDNA HS Assay Kit on a Qubit 3.0 Fluorometer (Life Technologies Europe BV, Bleiswijk, The Netherlands).

### 2.3. Genome Sequencing (Oxford Nanopore Technologies and Illumina)

The DNA from sample 18-2804 was used to prepare a 1D ligation library using the Ligation Sequencing Kit SQK-LSK110 according to the manufacturer’s instructions (Oxford Nanopore Technologies, Oxford, UK). ONT libraries were run on a PromethION flowcell (FLO-PRO002) at Future Genomics Technologies BV (Leiden, The Netherlands) using the following settings: basecall model: high-accuracy; basecaller version: Guppy v4.3.4.

Parallel aliquots of both DNA samples (18-2804 and 18-2837) were used to prepare Illumina libraries using the Nextera DNA Flex Library Prep Kit according to the manufacturer’s instructions (Illumina Inc. San Diego, CA, USA). Library quality was measured via electrophoresis in D1000 ScreenTape on an Agilent 4200 TapeStation System (Agilent Technologies Netherlands BV, Amstelveen, The Netherlands). The genomic paired-end (PE) libraries were sequenced with a read length of 2 × 150 nt using the Illumina NovaSeq 6000 system. Image analysis and basecalling were performed by the Illumina pipeline.

### 2.4. Genome Assembly and Annotation

Three reference genomes were used for the removal of host-derived ONT reads: *Arvicola amphibious* (GCA_903992535.1), *Apodemus sylvaticus* (GCA_001305905.1), and *Myodes glareolus* (GCA_004368595.1). Contigs were de novo assembled from the unaligned reads using Flye v2.8.3-b1695 in standard mode and in metagenome mode [30,31]. The “metagenome” contigs were further polished using Medaka v1.4.3 [32]. The filtered reads from the ONT data set were aligned against the Medaka-polished assembly, which were then subsequently used for a de novo assembly using Flye with polishing using Medaka (Figure 1).

Illumina reads of samples 18-2837 and 18-2804 were aligned against the ONT-based consensus sequence of *Ca.* N. mikurensis (1,236,636 bp; PromethION derived from spleen 18-2804) using minimap2 v 2.17. Pilon vs. 1.23 [33] was then used to polish the ONT-based consensus sequence of *Ca.* Neoehrlichia mikurensis (18-2804_Ehrlichia_flye_medaka_prokka.fna) independently with the 18-2837 and 18-2804 set of aligned Illumina reads. Prokka v1.14.6 [34] was used to annotate the polished genome sequences. BUSCO v5.2.2 [35] was used for QC of the annotated genome sequences based on the rickettsiales_odb10 lineage dataset. The sorted Illumina reads of 18-2837 and 18-2804 and the sorted nanopore reads of 18-2804 were aligned back against both polished genome sequences to allow visualization of the Bam/Bai files in IGV v2.12.2 [36]. The presence or absence of prophages was determined using the online tool Phaster [37,38]. Furthermore, SAPP [39] was used for the functional annotation of protein coding genes using InterProScan [40] with PFAM [41]. The web version of eggNOG-Mapper v2 [42,43] was used to determine the Clusters of Orthologous Group (COG) categories for protein encoding regions.

### 2.5. Pangenomic and Comparative Analyses

The Illumina reads of NL07 were mapped to SE20 using minimap2 v2.7 [44], and samtools v1.12 [45] was used to index the bam file and sibeliaZ [46] to generate a maf file. The maf file and bam files were visualized using IGV v2.12.2 and Tablet v1.17.08.17 [47,48], respectively.

The *Neoehrlichia* pangenome analysis was performed by following the anvi’o pangenomic workflow, and the mcl inflation was set to 2, using anvi’o v7 [49]. For the analyses, the genomes of *Ca.* N. mikurensis SE24, SE20, SE 26, *Ehrlichia chaffeensis* Arkansas, *Ehrlichia ruminantium* Welgevonden, strain *Anaplasma phagocytophilum* HZ, and *Ca.* Neoehrlichia lotoris RAC-413 were downloaded from NCBI (accessions numbers are listed in Appendix A). The Anvi’o genome databases were annotated using the NCBI COG function. A presence–absence Table was used to generate UpSet plots using the R package UpSetR [50] to visualize unique and shared gene clusters at both the intra- and interspecies levels.

### 2.6. Variant Calling

Single-nucleotide polymorphisms (SNPs) and other variants among the reference genome NL07, the Illumina reads of NL06, and the three published *Ca.* N. mikurensis genomes [18] were identified using Snippy v4.6.0 [51].

## 3. Results

From the 76 *M. glareolus* captured, 24 spleen samples tested positive for *Ca.* N. mikurensis DNA in the qPCR analysis (Appendix A). Four samples with the lowest Ct-values were selected for genomic DNA extraction and microbial enrichment using the NEBNext Microbiome DNA Enrichment Kit (New England BioLabs, Ipswich, MA, USA). The two samples with the highest DNA yield were subjected to genomic sequencing (sample 18-2804 and 18-2837).

### 3.1. Genomes Generated in This Study

Two complete and circular *Ca.* N. mikurensis genomes derived from mice spleen samples were assembled in this study. The reference genome derived from PromethION (sample 18-2804) was polished with Illumina data from the same sample, resulting in a circular genome referred to as NL07 (GenBank accession no. CP089285). In addition, the PromethION data derived from sample 18-2804 were polished with Illumina data from sample 18-2837, resulting in a second circular genome referred to as NL06 (GenBank accession no. CP089286). Both assemblies presented a complete genome with high BUSCO scores, which increased from 77.1% to >97% after the short reads were used to polish the long-read assembly and were accurately correct for sequence errors (Table 1, Appendix A). No prophages were identified in either genome.

The two genome assemblies generated in this study were compared for strain variations (including both indels and nucleotide substitutions), and in total, 250 variants were found (0.02% difference between genomes). Of these, 153 single nucleotide polymorphisms (SNPs), 34 insertions, 47 deletions, and 16 complex variants were detected. Out of these, two missense variants and three deletions were found in regions pertaining to the P44/Msp2 outer membrane protein (Appendix A).

As the consensus sequence NL07 presented a higher level of completeness (BUSCO = 99.2%), and both short and long reads were obtained from the same sample, it was used as our reference genome for all downstream analyses. This reference genome (NL07) has 949 coding sequences (CDS) out of 990 genes as well as 3 rRNAs and 37 tRNAs (Table 1). Coding proteins were classified into functional Clusters of Orthologous Group (COG) categories (Appendix A), and the output was summarized into the number of coding proteins belonging to each COG category (Appendix A).

### 3.2. Intraspecies Comparisons

NL07 was compared to three previously published genomes of *Ca.* N. mikurensis, namely strains SE20, SE24, and SE26, all of which were obtained directly from patient materials in Sweden [18]. Of notable importance, when compared to NL07, the published genomes were on average 124,574 bp smaller. The comparison showed SE26 is the most similar to NL07 (0.02% difference and 165 SNPs), and SE24 is the most distant (0.028% and 257 SNPs) (Table 2). Moreover, when compared to the published strains, NL07 shows mutations that could translate into phenotypic antigenic differences in the coding region of four P44/Msp2 outer membrane proteins (Table 3, Appendix A). In terms of completeness, the BUSCO scores of SE20, SE24, and SE26 are lower than NL07 (93.7%, 94%, 94.3%, and 99.2%, respectively, Appendix A), which suggests missing conserved genes in the previously published assemblies.

When investigating the 124,574 bp that were absent in the published genomes from patient samples, we found three main expansions that contain 31 genes belonging to 26 known protein domains as well as repeats of the outer membrane protein domain PF01617 (Appendix A, Appendix A). All but one of the missing genes were most closely related to *Ca.* Neoehrlichia lotoris. The remaining gene was most similar to a domain participating in biotin metabolism found in *Ehrlichia chaffeensis* (Appendix A). The Clusters of Orthologous Group (COG) categories assigned to these protein-coding genes are related to various essential processes needed for bacterial survival (Table 4), with the most abundant involved in replication, recombination, and repair (seven protein-coding domains) as well as translation, ribosomal structure, and biogenesis (five protein-coding domains).

The absent genes appear in three main gaps. Upon closer inspection, two of these gaps contain repeats of an outer membrane protein belonging to the Pfam PF01617 domain (Figure 2).

Mapping the Illumina reads of NL07 to the assembly of SE20 shows that many copies of this surface antigen are stacked on top of a site (positioned around 743,000–755,000 bp in SE20) (Appendix A). This may be indicative of a collapsed repeat of the surface antigen explaining part of the discrepancy between genome sizes. Moreover, the repeats of the PF01617 domain represent 25 different e-values ranging from 1.8 × 10^−6^ to 3 × 10^−74^ that may point to antigenic variation (Appendix A).

### 3.3. Pangenome Analysis

NL07 was compared to select genomes of the Anaplasmataceae family as well as the *Ca.* N. mikurensis strains from Sweden (Table 5). The GC content of our reference genome (26.85%) is comparable to that of the published strains (26.84%) and close to that of *E. ruminantium* and *Ca.* N. lotoris (27.48% and 27.75, respectively) that shares a similar genome size (Table 5, Figure 3). Four hundred and sixty-three gene clusters are present across all genomes (Figure 4). *Anaplasma phagocytophilum* has the largest genome, and 523 unique gene clusters (Figure 3 and Figure 4). In contrast, NL07 has 13 unique gene clusters and 13 that it only shares with the *Ca.* N. lotoris genome, the only other genome with which it solely shares gene clusters (Figure 3 and Figure 4). Among the shared clusters are one gene cluster connected to cell motility, two related to cell wall/membrane/envelope biogenesis of which one is an outer membrane protein, one connected to translation, ribosomal structure, and biogenesis, and one connected to inorganic ion transport and metabolism and a TPR-like repeat domain (Appendix A).

## 4. Discussion

Two novel and complete *Ca.* N. mikurensis genomes have been generated in this study using a reproducible approach for high-quality whole genome assembly directly from rodent spleens collected in the wild. These genomes expand on our ability to identify potential targets for the development of reliable diagnostic tools for neoehrlichiosis, which are currently lacking for this and some other tick-borne bacteria.

The genomes presented in this study are approximately 10% larger than the existing *Ca.* N. mikurensis assemblies recently published, which were derived from clinical samples [18]. The discrepancy in chromosome size might be related to genetic divergence rooted in the provenance of the samples or to the difference in technological platforms and assembly approaches employed.

Although the genes missing in the variants from Sweden are all involved in essential processes, one could argue the gene loss is related to pathogenicity gain as has been shown for other intracellular bacteria [52,53,54,55]. In order to investigate this hypothesis, a larger comparison of host- versus patient-derived genomes must be performed.

The genomes in this study are products of a hybrid assembly approach combining long and short reads, while the genomes from Sweden are based on short reads alone. While short reads are highly accurate at the nucleotide level, they lack the ability to reliably elucidate genome structure [23]. When mapped to the assembly of the Swedish variant SE20, our short-read data of NL07 revealed a large spike containing a repeat of an outer membrane protein domain, which has proven to be highly immunogenic in patients infected with other members of the family *Anaplasmataceae* [56]. Given that the Swedish variants were assembled based on short reads alone, it is possible that this domain, which appears throughout the genome, collapsed into one locus in said assemblies, explaining part of the discrepancy in genome sizes (Appendix A).

The relatively high copy number of this domain could be related to the adaptive immunogenic capabilities of *Ca.* N. mikurensis [57]. In *A. phagocytophilum*, this domain has over 113 copies, which has been associated with an increased adaptability to the environment during infection [58], a phenomenon that has been described in the surface protein superfamily (Pfam01617) for *A. marginale*, *E. canis*, *E. chaffeensis*, and *E. ruminantium* [59]. In *A. phagocytophilum*, p44/msp2 proteins present strain variability, which could explain why our analyses show SNPs in this domain, between the genomes generated in this study as well as between NL07 and the publicly available *Ca.* N. mikurensis genomes. Thus, we believe this surface protein family should be studied in depth in order to understand the evolutionary processes involved and how they affect antigenic variation for this potentially emerging pathogen.

The genomes presented in this study provide a foundation for future studies that could explore the antigenic variation of *Ca.* N. mikurensis. Moreover, we believe that this approach, in which wildlife reservoir host derived tissues are directly used to obtain high-quality whole genomes based on hybrid sequencing, should be employed for other emerging tick-borne pathogens and symbionts.

## Figures and Tables

**Figure 1 microorganisms-10-01134-f001:**
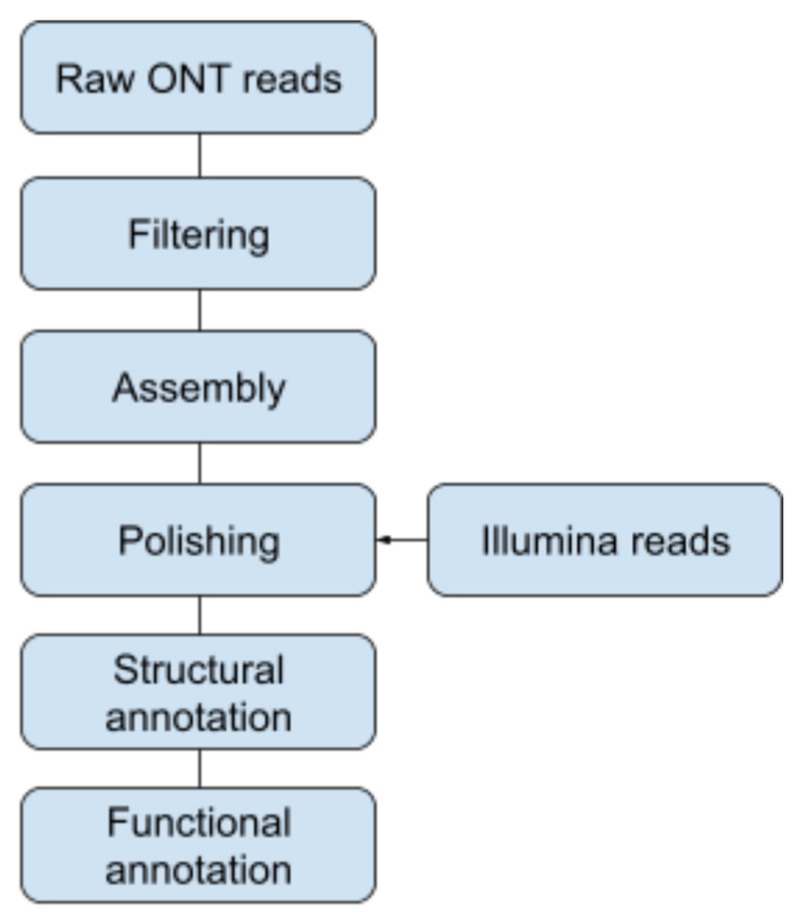
Overview of the assembly workflow. Raw ONT reads were filtered followed by a draft assembly. The assembly was curated using Illumina reads.

**Figure 2 microorganisms-10-01134-f002:**
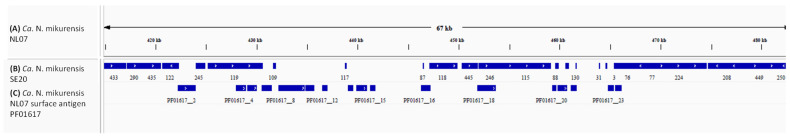
IGV plot indicating (**A**) the full assembly of NL07, (**B**) in blue, the regions of the SE20 assembly that align to NL07 and the gaps that SE20 does not encompass, and (**C**) in blue, the location of repeats of the outer membrane protein repeats belonging to the PF01617 domain in NL07 and SE20. Note that most repeats are present only in NL07.

**Figure 3 microorganisms-10-01134-f003:**
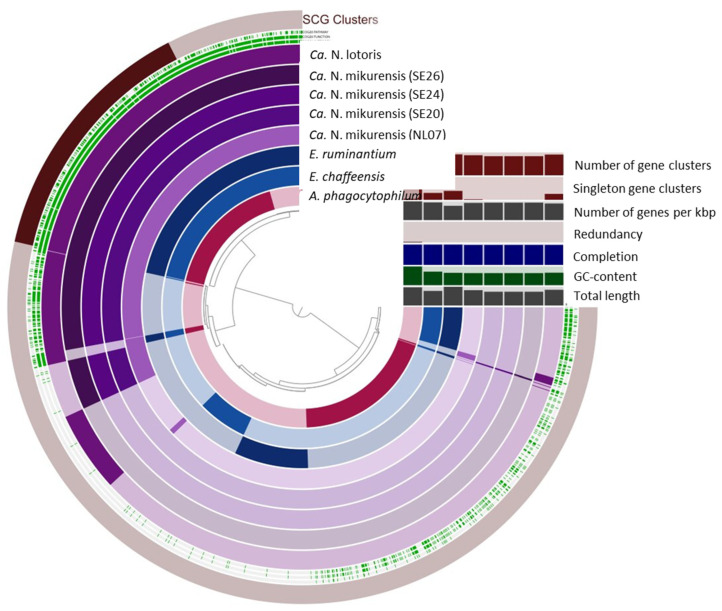
Comparative Anvi’o genomic analysis of NL07 and the additional *Ca.* N. mikurensis, *Ca.* N. lotoris, *E. chaffeensis*, *E. ruminantium*, and *A. phagocytophilum* genomes included in this study based on the presence/absence of gene clusters. The inner layers represent individual genomes organized by their phylogenetic relationships as indicated by the dendrogram. In the layers, dark colors indicate the presence of a gene group, and light colors indicate its absence. Number of gene clusters, singleton gene clusters, number of genes per kbp, redundancy, completion, GC content, and total length are represented in bar plots. SCG clusters = Single copy gene clusters.

**Figure 4 microorganisms-10-01134-f004:**
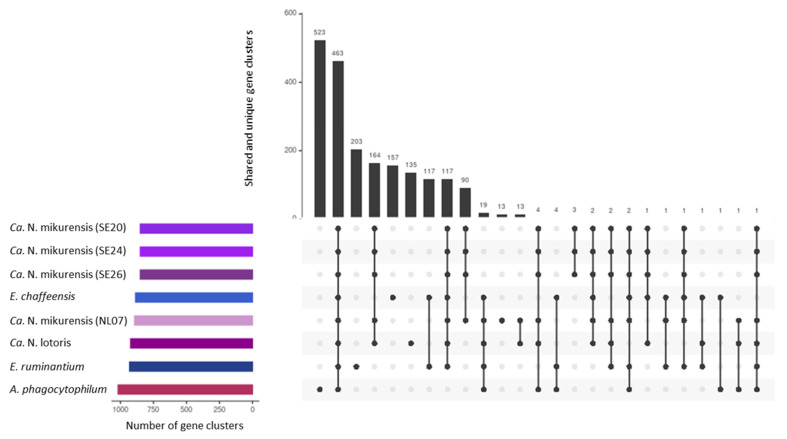
UpSet plot representing the shared and unique gene clusters between NL07 and other members of the Anaplasmataceae family. Black dots indicate the presence of a gene cluster, and connected dots indicate their presence across genomes. The colored horizontal bars represent the amount of gene clusters per genome ranging from 850 to 1018.

**Table 1 microorganisms-10-01134-t001:** Specifications of the unpolished and Illumina polished *Ca.* N. mikurensis consensus sequences. Note that NL07 was assembled based on PromethION + Illumina data from the same spleen sample (18-2804), while NL06 was assembled based on PromethION data from 18-2804 and Illumina data from 18-2837. All further analyses are based on NL07 only.

Sample Name	PromethION 18-2804	PromethION + Illumina 18-2804 (NL07)	PromethION 18-2804 + Illumina 18-2837 (NL06)
Assembly size	1,236,636	1,236,870	1,236,136
No. CDS	1152	949	958
No. gene	1193	990	999
No. rRNA	3	3	3
No. tRNA	37	37	37
BUSCO score	77.1%	99.2%	97.8%

**Table 2 microorganisms-10-01134-t002:** Genetic variation between NL07 and SE20 and SE24 and SE26. The Table shows the total amount of variants between NL07 and a given strain (Variant total), the number of multiple nucleotide polymorphisms (Complex), the number of deletions (Deletions), the number of insertions (Insertions), the number of single nucleotide polymorphisms (SNPs), the assembly size (Genome size), and the percentage in difference between NL07 and a given strain in the aligned regions (% difference).

Strain	Variant Total	Complex	Deletions	Insertions	SNPs	Genome Size (NL07 = 1,236,870)	% Difference
SE20	336	16	30	53	237	1,112,315	0.027
SE24	349	13	31	48	257	1,112,301	0.028
SE26	247	21	27	34	165	1,112,271	0.020

**Table 3 microorganisms-10-01134-t003:** P44/Msp2 family outer membrane protein variants between NL07 and *Ca.* N. mikurensis SE20, SE24, and SE20 assemblies. The effects of the SNPs are presented as synonymous (functionally silent) or nonsynonymous. Nonsynonymous variants, which lead to either a stop codon or a change in protein sequence, are in bold.

Strain	Type	Nucleotide Position	Effect
SE20	**complex**	**917/999**	**stop_gained c.917_919delTACinsAAT p.LeuLeu306**
	snp	258/903	synonymous_variant c.258C > T p.Pro86Pro
	snp	792/903	synonymous_variant c.792T > C p.Pro264Pro
	snp	168/852	synonymous_variant c.168G > A p.Pro56Pro
	**snp**	**287/852**	**missense_variant c.287G > A p.Ser96Asn**
	**snp**	**440/816**	**missense_variant c.440C > T p.Ala147Val**
SE24	**complex**	**917/999**	**stop_gained c.917_919delTACinsAAT p.LeuLeu306**
	snp	792/903	synonymous_variant c.792T > C p.Pro264Pro
	snp	168/852	synonymous_variant c.168G > A p.Pro56Pro
	**snp**	**287/852**	**missense_variant c.287G > A p.Ser96Asn**
	**snp**	**253/936**	**missense_variant c.253C > T p.Pro85Ser**
SE26	snp	552/903	synonymous_variant c.552A > G p.Gly184Gly
	snp	792/903	synonymous_variant c.792T > C p.Pro264Pro
	snp	168/852	synonymous_variant c.168G > A p.Pro56Pro
	**snp**	**287/852**	**missense_variant c.287G > A p.Ser96Asn**
	**snp**	**433/816**	**missense_variant c.433G > A p.Glu145Lys**

**Table 4 microorganisms-10-01134-t004:** Clusters of Orthologous Groups assigned to the protein-coding genes found in NL07 and missing in the published *Ca.* N. mikurensis genomes.

COG Categories	Description	Number of Genes
L	Replication, recombination, and repair	7
J	Translation, ribosomal structure, and biogenesis	5
H	Coenzyme transport and metabolism	3
C	Energy production and conversion	2
F	Nucleotide metabolism and transport	2
M	Cell wall/membrane/envelope biogenesis	2
P	Inorganic ion transport and metabolism	2
G	Carbohydrate metabolism and transport	1
T	Signal transduction mechanisms	1
U	Intracellular trafficking, secretion, and vesicular transport	1

**Table 5 microorganisms-10-01134-t005:** Summary of analyzed genomes.

Microorganism	Genome Length	GC Content	Gene Clusters	Singleton Gene Clusters
*A. phagocytophilum*	1,471,282	41.64	1018	523
*E. chaffeensis*	1,176,248	30.10	886	157
*E. ruminantium*	1,512,977	27.48	931	203
NL07	1,236,870	26.85	893	13
SE20	1,112,315	26.84	850	0
SE24	1,112,301	26.84	850	0
SE26	1,112,271	26.84	850	0
*Ca.* N. lotoris	1,268,660	27.75	923	135

## Data Availability

The genome sequences are present in GenBank under accession numbers CP089285 (NL07) and CP089286 (NL06).

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
