# Peer review of "Assembly and Comparison of Ca. Neoehrlichia mikurensis Genomes"

_microorganisms, 2022, doi:10.3390/microorganisms10061134_

Round 1
Reviewer 1 Report
The manuscript by Azagi et al., describes the genome sequencing and analysis of “Candidatus Neoehrlichia mikurensis”. The paper is potentially interesting, but needs some revision. They set out to get a genome sequence, and they got one!
Firstly, the authors have not used the correct name for the organism they have sequenced. According to the List of Prokaryotic Names with Standing in Nomenclature (https://lpsn.dsmz.de/genus/neoehrlichia), this organism has not been validly published, and should be called Candidatus. Please read about it!
Secondly, the authors claim to be providing the first full length genome sequences for this organism, however, the previously published CP054597, CP066557, and CP060793 all appear to be complete single contigs (Grankvist, et al., 2021), and the authors are aware, as they have cited this paper.
Lines 54-62: It is a point of debate as to whether mixing long and short read sequences is the best way to go. The short reads will often end a contig with incorrectly placed sequences that confound assembly, but I won’t belabor the point… but don’t try to tell me that is the thing to do. This will not get standardized – it will depend on the pathogen.
In the Anaplasmas msp2 genes tend to be dispersed throughout the genome. In the Ehlichia’s there tends to be a “sink” of msp2-type genes. In your supp fig 2 (your figure legend does not convey enough information), you indicate that msp4 stacks up against the Swedish sequences, suggesting 1) that the Swedish genomes are misassembled, and 2) msp4 is repeated in your genome. If there are a bunch of these genes they would stack up at any one place that had one in the assembly. However in the next paragraph, you go on to discuss msp2 being found in many copies in a bunch of organisms… you need to be careful – there are only a few in A. marginale, A. ovis, A. centrale (there are other genes that map to the 01617 pfam! Of which msp4 is one) The Ehrlichas have ~16-20. It is only A. phagocytophilum that has many msp2 genes. The A. platys genome paper was written in such a way that it was difficult to tell if they were actually speaking of msp2 or other members of the msp2 superfamily. If you do not have domain knowledge of these very important genes, you should get help from an expert with your annotation because it is easy to get confused! You do not have any (real) figures explaining your understanding of these genes, and you keep bringing up pfam01617, so it is not clear if you are distinguishing between msp2, msp4 and other members of the superfamily. If you really have 114 tandem copies of msp2 that would be interesting and unusual… but I would never say I know what the next bug in this batch is going to show me, because they each have a little twist!
Figure 2... Legend needs more explanation - the way it is drawn, it looks like the same gene compliment in both. All those blue genes are right next to "SE20". How am I supposed to interpret this?
Your supplementary files need an explanation of what they contain – there is no informative information given at the end of your paper (lines 328-331). Please explain what the numbers mean in Sup file 1-1. Sup File 1-2 – this is such a small amt of info, you can include in the text (no table necessary), ditto for Sup File 1-3 (As a small note, people do not think in RefSeq numbers, I would use the original annotation # unless you have a good reason for not doing that.)
Sup Table 3/Figure 3: The resolution of Fig 3 isn’t good. What is the point of Sup Fig 3? It seems a bit pointless to include EVERY genome seq that you can lay your hands on, especially once you are outside of the family Anaplasmataceae (it just makes your tree huge and doesn’t illustrate a point). But there is something VERY odd with your tree: The genus Ehrlichia groups more closely with Anaplasma phagocytophilum and A. platys than A. phagocytophilum and A. platys group with Anaplasma marginale. You cannot do these analyses using the whole genome… you need to do them using a common set of proteins – your methods are not detailed enough for me to understand exactly what you did, but knowing this family pretty well, yes, there is a deep clade between the livestock infecting Anaplasmas (A. marginale, A. centrale, A. ovis) and the zoonotic Anaplasmas (the Aps), but Ehrlichia is in a separate genus for a reason. I would pick a few and redo this in a way that makes sense.
Generally all figures and Tables need more explanation. Tables need titles. Help us understand what you are trying to show us.
Author Response
We thank the reviewer for bringing to our attention several issues that needed to be resolved in order for the manuscript to be more accurate in its focus and also more clear to the reader. We did our utmost to address all comments, and believe that thanks to this “Assembly and comparison of Ca. Neoehrlichia mikurensis genomes” is a better manuscript than it was prior to revision.
Comments and Suggestions for Authors
The manuscript by Azagi et al., describes the genome sequencing and analysis of “Candidatus Neoehrlichia mikurensis”. The paper is potentially interesting, but needs some revision. They set out to get a genome sequence, and they got one!
Firstly, the authors have not used the correct name for the organism they have sequenced. According to the List of Prokaryotic Names with Standing in Nomenclature (https://lpsn.dsmz.de/genus/isualizeia), this organism has not been validly published, and should be called Candidatus. Please read about it!
- We acknowledge this is the common nomenclature and have amended all instances in which “Neoehrlichia mikurensis” or “ mikurensis” is mentioned to read instead “Candidatus Neoehrlichia mikurensis” or “Ca. N. mikurensis”.
Secondly, the authors claim to be providing the first full length genome sequences for this organism, however, the previously published CP054597, CP066557, and CP060793 all appear to be complete single contigs (Grankvist, et al., 2021), and the authors are aware, as they have cited this paper.
- When writing “Here, we present the first complete Neoehrlichia mikurensis genomes directly derived from host tissues” we meant to convey these are the first complete Candidatus Neoehrlichia mikurensis genomes derived from reservoir hosts, aka wildlife, and have thus amended the text for clarity to read “Here, we present the first complete Neoehrlichia mikurensis genomes directly derived from wildlife reservoir host tissues …” (line 18) and “Moreover, we believe that this approach, in which wildlife reservoir host derived tissues…” (line 328).
Lines 54-62: It is a point of debate as to whether mixing long and short read sequences is the best way to go. The short reads will often end a contig with incorrectly placed sequences that confound assembly, but I won’t belabor the point… but don’t try to tell me that is the thing to do. This will not get standardized – it will depend on the pathogen.
- Mixing both short and long sequences has allowed us to assemble Ca. Neoehrlichia mikurensis genomes more accurately mainly because of the repetitive elements found in these genomes. We agree that this should not reflect on any assembly one should attempt but can be considered in similar situations. We have made the point more specific to our project by writing “Moreover, the use of hybrid assembly strategies, combining long and short read sequencing, is recommended for more complete and accurate reference genome assemblies especially when repetitive elements are expected” (lines 58-59).
In the Anaplasmas msp2 genes tend to be dispersed throughout the genome. In the Ehlichia’s there tends to be a “sink” of msp2-type genes. In your supp fig 2 (your figure legend does not convey enough information), you indicate that msp4 stacks up against the Swedish sequences, suggesting 1) that the Swedish genomes are misassembled, and 2) msp4 is repeated in your genome. If there are a bunch of these genes they would stack up at any one place that had one in the assembly.
- We now see the legend of supplementary figure 2 was not informative enough and it has now been amended, moreover the figure’s explanation now reads: “Neo_07 Illumina reads were aligned to the published Swedish genome SE20 using Minimap2. As visualized with Tablet, there is an accumulation of reads (A) encoding the PF01617 outer membrane domain that stack on top of the SE20 assembly (B) in the region around 743000-755000bp.” .
However in the next paragraph, you go on to discuss msp2 being found in many copies in a bunch of organisms… you need to be careful – there are only a few in A. marginale, A. ovis, A. centrale (there are other genes that map to the 01617 pfam! Of which msp4 is one) The Ehrlichas have ~16-20. It is only A. phagocytophilum that has many msp2 genes. The A. platys genome paper was written in such a way that it was difficult to tell if they were actually speaking of msp2 or other members of the msp2 superfamily. If you do not have domain knowledge of these very important genes, you should get help from an expert with your annotation because it is easy to get confused! You do not have any (real) figures explaining your understanding of these genes, and you keep bringing up pfam01617, so it is not clear if you are distinguishing between msp2, msp4 and other members of the superfamily. If you really have 114 tandem copies of msp2 that would be interesting and unusual… but I would never say I know what the next bug in this batch is going to show me, because they each have a little twist!
- We thank the reviewer for pointing out that indeed in order to refer to these outer membrane proteins by name we would need to perform a different type of analysis. Our analyses show we have copies of an outer membrane protein belonging to the PF01617 domain. We have now refrained from referring to the outer membrane proteins by name when talking about the repeats of Pfam PF01617 in the “Intra-species comparisons” section and in Figure 2 (lines 228, 242, 253).
Regarding the “Discussion” section, here we only refer to A. phagocytophilum having 113 copies of this domain : “In A. phagocytophilum, this domain has over 113 copies which has been associated with increased adaptability to the environment during infection [59],…” but we make no such claims regarding the Ca. N. mikurensis genomes. The comparison is made to highlight that multiple copies of the PF01617 domain have been reported in other members of the family Anaplasmataceae, where the phenomenon has been shown to have biological implications.
Figure 2... Legend needs more explanation – the way it is drawn, it looks like the same gene compliment in both. All those blue genes are right next to “SE20”. How am I supposed to interpret this?
- We thank the reviewers for bringing to our attention the lack of information in this figure, this has been amended in the figure’s legend, and by expanding upon the figure explanation to read : “IGV plot indicating (A) the full assembly of NL07, (B) in blue, the regions of the SE20 assembly that align to NL07 and the gaps that SE20 does not encompass, (C) in blue, the location of repeats of the outer membrane protein repeats belonging to the PF01617 domain in NL07 and SE20. Note that most repeats are present only in NL07.”.
Your supplementary files need an explanation of what they contain – there is no informative information given at the end of your paper (lines 328-331). Please explain what the numbers mean in Sup file 1-1. Sup File 1-2 – this is such a small amt of info, you can include in the text (no table necessary), ditto for Sup File 1-3 (As a small note, people do not think in RefSeq numbers, I would use the original annotation # unless you have a good reason for not doing that.)
- The supplementary tables have been re-named in a more informative way as well as supplementary files 1-5. Moreover the “Supplementary Materials” section has been re-writen to read:
“Supplementary Materials: The following supporting information can be downloaded at: www.mdpi.com/xxx/s1, Supplementary Tables 1-13 (1_qPCR results of spleen samples, 2_Primers, 3_External genomes, 4_SNPs_NL07-NL06, 5_Eggnog_output_NL07, 6_Eggnog_COGs_NL07, 7_SNPs_NL07-SE20, 8_SNPs_NL07-SE24, 9_SNPs_NL07-SE26, 10_Genes NL07 only, 11_Eggnog_ouput_GAP, 12_PF01617 repeats, 13_Anvio output), title: Supplementary_File 1_tables; Supplementary Files 1-5 (Supplementary_file_1_BUSCO_NL06_CP089286, Supplementary_file_2_BUSCO_NL07_CP089285, Supplementary_file_3_BUSCO_SE20_CP054597, Supplementary_file_4_BUSCO_SE24_CP066557, Supplementary_file_5_BUSCO_SE26_CP060793),title: Supplementary_Files_1-5, Supplementary Figures 1,2 (Supplementary figure 1, IGV output showing gaps in SE20; Supplementary figure 2, tablet visualization showing accumulation of PF01617 repeats) , title: Supplementary figure 1_2.”
Sup Table 3/Figure 3: The resolution of Fig 3 isn’t good. What is the point of Sup Fig 3? It seems a bit pointless to include EVERY genome seq that you can lay your hands on, especially once you are outside of the family Anaplasmataceae (it just makes your tree huge and doesn’t illustrate a point). But there is something VERY odd with your tree: The genus Ehrlichia groups more closely with Anaplasma phagocytophilum and A. platys than A. phagocytophilum and A. platys group with Anaplasma marginale. You cannot do these analyses using the whole genome… you need to do them using a common set of proteins – your methods are not detailed enough for me to understand exactly what you did, but knowing this family pretty well, yes, there is a deep clade between the livestock infecting Anaplasmas (A. marginale, A. centrale, A. ovis) and the zoonotic Anaplasmas (the Aps), but Ehrlichia is in a separate genus for a reason. I would pick a few and redo this in a way that makes sense.
- In light of the remarks of reviewers 1 and 2 and because a comparison between the genomes generated in this study and other relevant members of the family Anaplasmataceae is already shown in the pangenome analysis, we have decided to remove the phylogenetic analysis and related figures, including Figure 3 and Supplementary Figure 3. Figure numbers have been corrected accordingly.
Generally all figures and Tables need more explanation. Tables need titles. Help us understand what you are trying to show us.
- Further changes have been made to figures 1, 4 and 5 (now figures 3 and 4) as well as Tables 1-3.
Reviewer 2 Report
The manuscript by Azagi et al. presents the assembly and comparison of two Neoehrlichia mikurensis genomes. The work is important and the methodology used is sound. However, the presentation of the results is rather poor. In addition to my comment below, I really encourage the authors to improve the figures and the presentation of the results. Genome comparison results presentation used in some of the classic papers in the field (e.g., PLoS Genet. 2006;2(2):e21. doi: 10.1371/journal.pgen.0020021.) can be used as guidance.
Introduction
This section can include citation to some of the literature on genomics applied to the study of bacteria of the family Anaplasmataceae.
Methods
Line 157: ‘N. mikurensis’ in Italics.
Results
Line 171: Remove ‘(26,3<Ct<28,0)’
Line 186-187: This does not make sense: ‘compared for strain variations’ and ‘in total 250 variants’ reads as if ‘250 strain variants’ were found. I guess the authors are referring to ‘250 nucleotide substitutions’ were found when comparing the two genomes (?). Please, revise and correct.
Line 247: The Pangenome analysis could include some Venn Diagram showing the number of shared and species-specific genes in the genomes being compared.
Reference to supplementary figures in the text
Can the authors refer to one ‘supplementary file’ at the time (e.g., lines 184-185, line 196, and line 211). Also, correct ‘Supplemental’ to ‘Supplementary’ (lines 159-160).
Tables
Table 1. The results are not consistently presented. If two genomes were obtained, ‘NL07’ and ‘NL06’ (both using Nanopore and Illumina), why table 1 only has three columns. Following the logic of the table, shouldn’t each genome generate two columns (one for the results using Nanopore and the other one for the results using Nanopore-Illumina). Please, revise this. Independently, I advise to add only the data of the finally-assembled genome. In addition, try to use the nomenclature consistently throughout the text. Species name goes in Italics.
Table 3. Not clear what this table is showing. Some information should be added for the reader to understand what is for example ‘complex’ or ‘Effect’. Not clear what is shown under the column ‘Effect’. Not clear why ‘nucleotide substitutions’ (which can be ‘synonymous’ or ‘non-synonymous’) are being called ‘variants’. It is confusing. Please, correct this here and in the text.
Table 4. In the heading, ‘summary’ goes with capital ‘S’.
Figures
Figure 1. The legend needs revision. Figures have to be self-explanatory. In this case, several things are shown in the figure that are not explained. Also, the authors mentioned that they used Illumina and Nanopore sequencing, but from this ‘workflow’ is not clear how these two technologies were used in combination to get highly-quality genomes.
Figure 2 is poorly presented. It is not clear where is the gene msp2 (gene names go in Italics). Please, use arrows to help the reader visualise what is being shown. Increase the quality of the figure, it looks like a copy-paste from another screen, which in by itself is not problem as long as the figure has quality and this one does not have it.
Figure 3 is poorly presented. The quality of the phylogenetic tree is very low. Increase the quality of the figure, it looks like a copy-paste from another screen, which in by itself is not problem as long as the figure has quality and this one does not have it. Some of the branches were even cut (it is not enough to say that ‘The full tree is in Supplementary figure 3.’). Species name goes in Italics. There are several genomes of different members of the family Anaplasmataceae. Please, represent more of those in the tree (e.g., Other Anaplasma and Ehrlichia species for which genomes are available).
Figure 4. Some of the font size in the figure are very small. The strains of which genomes are included are not consistently named (e.g., one strain is named ‘Neolot’, the other ‘Neo SE26’, the other ‘Erum’ and so on). Please, be consistent with the naming of the strains (this applies also to the text of the manuscript).
Figure 5. No clear what the coloured ribbons represent here. Some of the font size in the figure are very small.
Author Response
We thank the reviewer for bringing to our attention several issues that needed to be resolved in order for the manuscript to be more accurate in its focus and also more clear to the reader. We did our utmost to address all comments, and believe that thanks to this “Assembly and comparison of Ca. Neoehrlichia mikurensis genomes” is a better manuscript than it was prior to revision.
Comments and Suggestions for Authors
The manuscript by Azagi et al. presents the assembly and comparison of two Neoehrlichia mikurensis genomes. The work is important and the methodology used is sound. However, the presentation of the results is rather poor. In addition to my comment below, I really encourage the authors to improve the figures and the presentation of the results. Genome comparison results presentation used in some of the classic papers in the field (e.g., PloS Genet. 2006;2(2):e21. Doi: 10.1371/journal.pgen.0020021.) can be used as guidance.
Introduction
This section can include citation to some of the literature on genomics applied to the study of bacteria of the family Anaplasmataceae.
- We appreciate this suggestion and now refer to two publications (Pubmed IDs 30665414, 35114560) in which genomes of the family Anaplasmataceae were assembled using a hybrid sequencing approach : “This approach has already been used for other members of the Anaplasmataceae in order to assemble high resolution whole genomes [26,27].” (lines 59-61).
Methods
Line 157: ‘N. mikurensis’ in Italics.
- This has been corrected.
Results
Line 171: Remove ‘(26,3<Ct<28,0)’
- We understand these values are superfluous and have deleted them from the “Results” section.
Line 186-187: This does not make sense: ‘compared for strain variations’ and ‘in total 250 variants’ reads as if ‘250 strain variants’ were found. I guess the authors are referring to ‘250 nucleotide substitutions’ were found when comparing the two genomes (?). Please, revise and correct.
- This comment has been taken into account, we have clarified what we mean by strain variants in the text (lines 180-181): “The two genome assemblies generated in this study were compared for strain variations (including both indels and nucleotide substitutions), and in total 250 variants were found…”. Moreover the explanation of Table 2 has been edited to give more information about the different types of variants and now reads : “The Table shows the total amount of variants between NL07 and a given strain (Variant total), the number of multiple nucleotide polymorphisms (Complex), the number of deletions (Deletions) , the number of insertions (Insertions), the number of single nucleotide polymorphisms (SNPs), the assembly size (Genome size) and the percentage in difference between NL07 and a given strain in the aligned regions (% difference).”.
Line 247: The Pangenome analysis could include some Venn Diagram showing the number of shared and species-specific genes in the genomes being compared.
- We acknowledge that a Venn diagram can nicely represent genomic comparisons. However, as we compared eight genomes a Venn diagram would not be informative and that’s why an Upset plot was chosen instead (Figure 4, previously figure 5) with referral to shared gene clusters in the text (lines 262-268).
Reference to supplementary figures in the text
Can the authors refer to one ‘supplementary file’ at the time (e.g., lines 184-185, line 196, and line 211). Also, correct ‘Supplemental’ to ‘Supplementary’ (lines 159-160).
- Lines 184-185 (178-179): we refer to table 1, but also to supplementary files 1 and 2 as they contain the complete BUSCO score reports for readers who wish to glean more information.
- Line 196 (191-193): has been redacted and now reads: “Coding proteins were classified into functional Clusters of Orthologous Groups (COG) categories (Supplementary Table 5) and the output was summarized into the number of coding proteins belonging to each COG category (Supplementary Table 6).”.
- Line 211: we refer to supplementary files 2-5 as they contain the complete BUSCO score reports for readers who wish to glean more information.
- Lines 159-160: this has been corrected.
Tables
Table 1. The results are not consistently presented. If two genomes were obtained, ‘NL07’ and ‘NL06’ (both using Nanopore and Illumina), why table 1 only has three columns. Following the logic of the table, shouldn’t each genome generate two columns (one for the results using Nanopore and the other one for the results using Nanopore-Illumina). Please, revise this. Independently, I advise to add only the data of the finally-assembled genome. In addition, try to use the nomenclature consistently throughout the text. Species name goes in Italics.
- We thank the reviewer for pointing out that the table is not clear, and have now amended the table description to read: “Table 1: Specifications of the unpolished and Illumina polished Ca. N. mikurensis consensus sequences. Note that NL07 was assembled based on PromethION + Illumina data from the same spleen sample (18-2804) while NL06 was assembled based on PromethION data from 18-2804 and Illumina data from 18-2837. All further analyses are based on NL07 only”. As it’s explained in lines 171-176 the assembly of NL07 is based on short and long reads from sample 18-2804, while the assembly of NL06 is based on long reads of sample 18-2804 and short reads of sample 18-2837. This, and the fact that the BUSCO score of NL07 is higher (99,2%) is the reason why NL07 is the chosen genome assembly for all downstream analyses (lines 186-188).
Table 3. Not clear what this table is showing. Some information should be added for the reader to understand what is for example ‘complex’ or ‘Effect’. Not clear what is shown under the column ‘Effect’. Not clear why ‘nucleotide substitutions’ (which can be ‘synonymous’ or ‘non-synonymous’) are being called ‘variants’. It is confusing. Please, correct this here and in the text.
- The table description has been edited so it can convey the missing information: “Table 3: P44/Msp2 family outer membrane protein variants between NL07 and Ca. N. mikurensis SE20, SE24 and SE20 assemblies. The effects of the SNPs are presented as synonymous (functionally silent) or non-synonymous. Non-synonymous variants, which lead to either a stop codon or a change in protein sequence, are in bold.”
Table 4. In the heading, ‘summary’ goes with capital ‘S’.
- This has been corrected.
Figures
Figure 1. The legend needs revision. Figures have to be self-explanatory. In this case, several things are shown in the figure that are not explained. Also, the authors mentioned that they used Illumina and Nanopore sequencing, but from this ‘workflow’ is not clear how these two technologies were used in combination to get highly-quality genomes.
- The Figure has been edited, now showing the workflow in a clearer manner that includes the polishing step using Illumina reads. The Figure explanation now reads: “Overview of the assembly workflow. Raw ONT reads were filtered followed by a draft assembly. The assembly was curated using Illumina reads.”
Figure 2 is poorly presented. It is not clear where is the gene msp2 (gene names go in Italics). Please, use arrows to help the reader isualize what is being shown. Increase the quality of the figure, it looks like a copy-paste from another screen, which in by itself is not problem as long as the figure has quality and this one does not have it.
- We thank the reviewer for bringing to our attention the lack of information in this figure, this has been amended in the figure’s legend, and by expanding upon the figure explanation to read : “IGV plot indicating (A) the full assembly of NL07, (B) in blue, the regions of the SE20 assembly that align to NL07 and the gaps that SE20 does not encompass, (C) in blue, the location of repeats of the outer membrane protein repeats belonging to the PF01617 domain in NL07 and SE20. Note that most repeats are present only in NL07.”.
Figure 3 is poorly presented. The quality of the phylogenetic tree is very low. Increase the quality of the figure, it looks like a copy-paste from another screen, which in by itself is not problem as long as the figure has quality and this one does not have it. Some of the branches were even cut (it is not enough to say that ‘The full tree is in Supplementary figure 3.’). Species name goes in Italics. There are several genomes of different members of the family Anaplasmataceae. Please, represent more of those in the tree (e.g., Other Anaplasma and Ehrlichia species for which genomes are available).
- In light of the remarks of reviewers 1 and 2 and because a comparison between the genomes generated in this study and other relevant members of the family Anaplasmataceae is already shown in the pangenome analysis, we have decided to remove the phylogenetic analysis, including figure 3 and supplementary figure 3. Figure numbers have been corrected accordingly.
Figure 4. Some of the font size in the figure are very small. The strains of which genomes are included are not consistently named (e.g., one strain is named ‘Neolot’, the other ‘Neo SE26’, the other ‘Erum’ and so on). Please, be consistent with the naming of the strains (this applies also to the text of the manuscript).
- The font size of the figure legends in Figure 4 (now Figure 3) has been increased, the genomes have been renamed and the figure description has been expanded upon for clarity: “Figure 3: Comparative Anvi'o genomic analysis of NL07 and the additional Ca. N. mikurensis, Ca. N. lotoris, chaffeensis, E. ruminantium and A. phagocytophilum genomes included in this study based on the presence/absence of gene clusters. The inner layers represent individual genomes organized regarding their phylogenetic relationships as indicated by the dendrogram. In the layers, dark colors indicate the presence of a gene group and light color its absence. Number of gene clusters, singleton gene clusters, number of genes per kbp, redundancy, completion, GC content and total length are represented in bar plots. SCG clusters = Single copy gene clusters.”
Figure 5. No clear what the coloured ribbons represent here. Some of the font size in the figure are very small.
- The font size of the figure legends in Figure 5 (now Figure 4) has been increased and the genomes have been renamed. Moreover an explanation regarding the colored ribbons has been added: “Figure 5: Upset plot representing the shared and unique gene clusters between NL07 and other members of the Anaplasmataceae family. Black dots indicate the presence of a gene cluster, connected dots indicate their presence across genomes. The colored horizontal bars represent the amount of gene clusters per genome ranging from 850 to 1018”.
Reviewer 3 Report
This menu is suitable for microorganisms.
Author Response
We thank the reviewer for taking the time to read and review our manuscript.
Round 2
Reviewer 2 Report
The authors have addressed all my previous concerns nicely. I recommend publication.